# Electromagnetic Conversion into Kinetic and Thermal Energies

**DOI:** 10.3390/e25091270

**Published:** 2023-08-29

**Authors:** Axel Brandenburg, Nousaba Nasrin Protiti

**Affiliations:** 1Nordita, KTH Royal Institute of Technology and Stockholm University, Hannes Alfvéns väg 12, 10691 Stockholm, Sweden; nopr1532@student.su.se; 2Oskar Klein Centre, Department of Astronomy, Stockholm University, AlbaNova, 10691 Stockholm, Sweden; 3School of Natural Sciences and Medicine, Ilia State University, 0194 Tbilisi, Georgia; 4McWilliams Center for Cosmology, Department of Physics, Carnegie Mellon University, Pittsburgh, PA 15213, USA

**Keywords:** electric energy, cosmological inflation, emergence of conductivity

## Abstract

The conversion of electromagnetic energy into magnetohydrodynamic energy occurs when the electric conductivity changes from negligible to finite values. This process is relevant during the epoch of reheating in the early universe at the end of inflation and before the emergence of the radiation-dominated era. We find that the conversion into kinetic and thermal energies is primarily the result of electric energy dissipation, while magnetic energy only plays a secondary role in this process. This means that since electric energy dominates over magnetic energy during inflation and reheating, significant amounts of electric energy can be converted into magnetohydrodynamic energy when conductivity emerges before the relevant length scales become stable.

## 1. Introduction

In hydrodynamic turbulence, a dissipation of energy is, in principle, straightforward; it must be equal to the energy input accomplished via forcing (see Figure 1). But, when magnetic fields are involved, energy can be transferred from kinetic energy to magnetic by working against the Lorentz force, $W_{L}$. In this case, the situation is more complicated because there are now two exit channels, and it is not clear a priori which of the two takes the lion’s share in specific situations (see Figure 2). A related problem may also occur when the electric energy reservoir is involved, especially when this energy reservoir is later absent due to high conductivity. Before discussing this, however, let us first recall the different situations in which hydrodynamic and hydromagnetic turbulence occur.

In Figure 2, it is presumed that kinetic energy can be tapped via dynamo action and converted into magnetic energy [1]. This is a generic process that we now know works in virtually all types of turbulent systems (provided the electric conductivity is large enough [2]). Large conductivity means small magnetic diffusivity; therefore, this also means less dissipation [3]. As shown in Figure 2, however, this seems puzzling; in the steady state, the dynamo term $W_{L}$ must be just as large as the resistive term, $\epsilon_{M}$. Thus, if the dynamo is efficient, then the dissipation must also be large, which is not expected (and also not true).

The puzzle of efficient dynamo action but inefficient dissipation was solved by realizing that, at large conductivities (especially when there is a large magnetic Prandtl number, which is the ratio ${\mathrm{Pr}}_{M}\equiv \nu /\eta$ of the kinematic viscosity ν to magnetic diffusivity η), a second conversion occurs at smaller length scales where magnetic energy can be converted back into kinetic energy. This process was termed a reversed dynamo [4], and it happens at small scales when ${\mathrm{Pr}}_{M}\gg 1$. The concept of a reversed dynamo was already introduced previously in the context of large-scale flows in two-fluid systems driven by microscopic fields and flows [5]. However, in the context of Ref. [4], the focus was on small-scale dynamos that drive small-scale flows via the Lorentz force when ${\mathrm{Pr}}_{M}\gg 1$.

While the conversion between magnetic and kinetic energies is reasonably well understood, not much is known about the conversion from electromagnetic energy, i.e., the sum of electric and magnetic energies, into magnetic and hydrodynamic energies when the electric conductivity gradually increases. Such a process is important at the end of cosmological inflation [6]. A stochastic electromagnetic field may have been produced during inflation and reheating [7]. At the end of reheating, the electric conductivity of the universe increased. As discussed in Ref. [8], significant magnetic field losses can occur if the increase in conductivity is slow, especially when the magnetic diffusivity is at an intermediate level for a long time. In the two extreme cases of very large diffusivity (which corresponds to a vacuum with undamped electromagnetic waves), and very small magnetic diffusivity (which corresponds to nearly perfect conductivity), no significant losses are expected. It is only during the period when the magnetic diffusivity is at an intermediate level that significant resistive losses can occur.

Once the conductivity has reached large values, i.e., when the magnetic diffusivity is small, strong turbulent flows will be driven. In that regime, the Faraday displacement current can be neglected, and the equations reduce to those of magnetohydrodynamics [9]. The resulting turbulent flows cause the magnetic field to undergo turbulent decay with inverse cascading, as studied intensively since the mid 1990s [10,11,12,13,14,15,16]. At some point around the time of recombination, the photon mean free path becomes very large, and a process called Silk damping becomes important [17]. This results from the interactions between photons and the gas, and all the inhomogeneities in the photon–baryon plasma are damped out [18]. In Ref. [19], this was modeled as a strongly increased viscosity, thereby making the magnetic Prandtl number even larger. However, a more physical approach is to add a friction term of the form −u/τ on the right hand side of the momentum equation [12]. It is generally taken for granted that magnetic fields just survive Silk damping without much additional loss and that they are simply frozen into the plasma. However, the details of this process have not yet been modeled. It is clear, however, that the assumption of a well-conducting universe is an excellent one, even after the epoch of recombination some 380,000 years after the Big Bang (when there were very few charged particles). As we emphasize below, the electric conductivity was then still large enough that the electric field was negligible, even in the voids between galaxy clusters. In cosmology, it is only near the end of inflation that the electric field can play a significant role. The electric energy density was then comparable to or in excess of the magnetic energy density.

The goal here is to understand more quantitatively how much magnetic energy survives during the conversions from electromagnetic fields to magnetohydrodynamic fields as the conductivity increases. We also consider, in more detail, the conversion from magnetic fields to electric fields at the end of the cosmological reheating phase, which is when both fields are still growing and not yet equal to each other, unlike the situation when electromagnetic waves are already established, and there is no longer any growth.

## 2. Energetics during the Emergence of Conductivity

The evolution of the electric and magnetic fields, E and B, respectively, is given in the Maxwell equations, written here in SI units as follows:(1)$$\frac{1}{c^{2}}\frac{\partial E}{\partial t}=\nabla \times B-\mu_{0}J,\;\nabla \cdot E=\rho_{e}/\epsilon_{0},$$
(2)$$\frac{\partial B}{\partial t}=-\nabla \times E,\;\nabla \cdot B=0,$$
where *c* is the speed of light, $\mu_{0}$ is the vacuum permeability, $\epsilon_{0}\equiv 1/\left(\mu_{0}c^{2}\right)$ is the vacuum permittivity, and $\rho_{e}$ is the charge density. To close the equations, we use Ohm’s law,
(3)J=σ(E+u×B),
where σ is the electric conductivity, and u is the velocity.

In the very early universe, inflation dilutes the plasma to the extent that there are virtually no particles, and hence, the electric conductivity vanishes. Eventually, a phase of reheating must have occurred. One possibility is that the stretching associated with the cosmological expansion leads to electromagnetic field amplification until the electric field begins to exceed the critical field strength for the Schwinger effect [20] to lead to the production of charged particles and, thus, to the emergence of electric conductivity. This change in σ implies the existence of a phase when σ has an intermediate value for a certain duration. This leads to a certain electromagnetic energy loss given as J·E. This is a well-known result in magnetohydrodynamics, where the displacement current is ignored, so we have $\nabla \times B=\mu_{0}J$. This is then used when deriving the magnetic energy equation by taking the dot product of Equation (Equation 2) with B, so we have
(4)$$\frac{\partial }{\partial t}\left(B^{2}/2\mu_{0}\right)=-B\cdot \nabla \times E/\mu_{0}=J\cdot E-\nabla \cdot \left(E\times B/\mu_{0}\right),$$
where we have introduced the Poynting vector $E\times B/\mu_{0}$. In the following, we often adopt volume averaging, which we denote in angle brackets. They depend just on time *t* but not on position x. We also adopt periodic boundary conditions in all three directions, so we call the domain triply periodic. Since a divergence under triply periodic volume averaging vanishes, we just have
(5)$$\frac{d}{dt}\langle B^{2}/2\mu_{0}\rangle =-\langle J\cdot E\rangle \;\left(\mathrm{ignoring}\;\mathrm{the}\;\mathrm{displacement}\;\mathrm{current}\right).$$
The 〈J·E〉 term, in turn, has two contributions. Using Ohm’s law in the form of
(6)E=J/σ−u×B,
we find $\langle J\cdot E\rangle =\langle J^{2}/\sigma \rangle -\langle J\cdot \left(u\times B\right)\rangle$, or, using −J·(u×B)=u·(J×B), we have
(7)$$\langle J\cdot E\rangle =\langle J^{2}/\sigma \rangle +\langle u\cdot \left(J\times B\right)\rangle ,$$
so part of the electromagnetic energy turns into Joule (or magnetic) heating, $\epsilon_{M}\equiv \langle J^{2}/\sigma \rangle$, and another part is converted into kinetic energy via work done by the Lorentz force, $W_{L}\equiv \langle u\cdot \left(J\times B\right)\rangle$, which eventually also becomes converted into heat through viscous (kinetic) heating, $\epsilon_{K}$. In the case of dynamo action discussed in the introduction, of course, $W_{L}$ is negative, so work is done against the Lorentz force. This is why the direction of the arrow in Figure 2 is reversed. Force-free magnetic fields have $W_{L}=0$ and can, therefore, not be sustained against dissipation, but they can be long lived if the current density is small enough; see Ref. [21] for examples.

In the scenario where reheating is caused by the feedback from the Schwinger effect, there would be thermal energy supply both from $\epsilon_{K}$ and $\epsilon_{M}$, leading, therefore, to a direct coupling between the resulting heating and the emergence of σ. The flows of energy between magnetic, electric, and kinetic energy reservoirs is illustrated in Figure 3. We denote those as
(8)$$E_{M}\equiv \langle B^{2}/2\mu_{0}\rangle ,\;E_{E}\equiv \langle \epsilon_{0}E^{2}/2\rangle ,\;and\;E_{K}\equiv \langle \rho u^{2}/2\rangle ,$$
respectively. Their evolution equations can be obtained from Equations (Equation 1) and (Equation 2), along with the momentum and continuity equations,
(9)$$\rho \frac{Du}{Dt}=-\nabla p+J\times B+\nabla \cdot \left(2\rho \nu S\right),$$
(10)$$\frac{D\ln\rho }{Dt}=-\nabla \cdot u,$$
where D/Dt≡∂/∂t+u·∇ is the advective derivative, $p=\rho c_{s}^{2}$ is the pressure for an isothermal equation of state with sound speed $c_{s}$, which is constant, ν is the viscosity, and $S_{ij}=\left(\partial_{i}u_{j}+\partial_{j}u_{i}\right)/2-\delta_{ij}\nabla \cdot u/3$ are the components of the rate-of-strain tensor S.

Note that, unlike the cases depicted in Figure 1 and Figure 2, there is no energy input in the system shown in Figure 3. This would change if we were to add forcing in the momentum equation in Equation (Equation 9). We allude to this interesting possibility at the end of the conclusions in Section 5. Another possibility that we do discuss in some detail is energy input during the reheating phase at the end of inflation. We turn to this aspect in Section 4.5.

Taking the dot product of Equation (Equation 9) with u, using Equation (Equation 10), integration by part, and the facts that $\partial_{i}u_{j}$ can be written as the sum of a symmetric and an antisymmetric tensor but that the multiplication with S (a symmetric and trace-free tensor) gives no contribution when $\delta_{ij}\nabla \cdot u/3$ is added, we find that $S_{ij}\partial_{i}u_{j}=S{}^{2}$ and, thus, obtain the evolution equation for the kinetic energy in the form
(11)$$\frac{d}{dt}\langle \rho u^{2}/2\rangle =-\langle u\cdot \nabla p\rangle +\langle u\cdot (J\times B)\rangle -\langle 2\rho \nu S{}^{2}\rangle ,$$
which we can also write more compactly as ${\dot{E}}_{K}=W_{P}+W_{L}-\epsilon_{K}$, where $W_{P}=-\langle u\cdot \nabla p\rangle$ has been defined as the work done by the pressure force, $\epsilon_{K}=\langle 2\rho \nu S{}^{2}\rangle$ is the viscous heating, and the dot on the kinetic energy $E_{K}$ (not to be confused with $\epsilon_{K}$) denotes a time derivative. Here, we have made use of the fact that the divergence ∇·(pu)=u·∇p+p∇·u has a vanishing volume average for a triply periodic domain and, therefore, −〈u·∇p〉=〈p∇·u〉, making it clear that this term leads to compressional heating and was found to be important in gravitational collapse simulations [22]. We will see later that when energy is supplied through $W_{L}$, the energy is used to let the kinetic energy grow (${\dot{E}}_{K}>0$) and to drive viscous heating, i.e., we have
(12)$$W_{L}={\dot{E}}_{K}+\epsilon_{K}-W_{P}.$$
We recall that the dot on $E_{K}$ denotes a time derivative. The term $W_{P}$ is usually small and negative and thus also contributes (but only little) to increasing thermal energy. In the present simulations, we used an isothermal equation of state and thus ignored the evolution of thermal energy, $E_{T}=\langle \rho e\rangle$, where $e=c_{v}T$ is the internal energy, $c_{v}$ is the specific heat at constant volume, and *T* is the temperature. If we had included it, we would have had
(13)$${\dot{E}}_{T}=\epsilon_{M}+\epsilon_{K}-W_{P}.$$
This thermal evolution is important in the simulations of thermal magneto-convection [23], where it facilitates buoyancy variations, or in the simulations of the magneto-rotational instability, where potential energy is converted into kinetic and magnetic energies that then dissipate as heat and radiation [24]. For our purposes, however, it suffices to integrate instead the kinetic and magnetic contributions in time, i.e., to compute $\int \epsilon_{K}\;dt$ and $\int \epsilon_{M}\;dt$, respectively.

Let us now discuss the interplay between electric and magnetic energies. This interplay is usually ignored in magnetohydrodynamics, where the evolution of the electric field, i.e., the Faraday displacement current, is ignored [9]. Taking the dot product of Equation (Equation 1) with $E/\mu_{0}$ and using $1/\left(\mu_{0}c^{2}\right)=\epsilon_{0}$, we obtain
(14)$$\frac{\partial }{\partial t}\left(\epsilon_{0}E^{2}/2\right)=\frac{E}{\mu_{0}c^{2}}\cdot \frac{\partial E}{\partial t}=E\cdot \nabla \times B/\mu_{0}-J\cdot E,$$
so, after averaging, we have
(15)$$\frac{d}{dt}\langle \epsilon_{0}E^{2}/2\rangle =\langle E\cdot \nabla \times B/\mu_{0}\rangle -\langle J\cdot E\rangle .$$
Next, taking the dot product of Equation (Equation 2) with $B/\mu_{0}$, we obtain
(16)$$\frac{\partial }{\partial t}\left(B^{2}/2\mu_{0}\right)=-B\cdot \nabla \times E/\mu_{0}.$$
In view of the $\langle E\cdot \nabla \times B/\mu_{0}\rangle$ term in Equation (Equation 15), it is convenient to rewrite Equation (Equation 16) in the form of
(17)$$\frac{\partial }{\partial t}\left(B^{2}/2\mu_{0}\right)=-E\cdot \nabla \times B/\mu_{0}-\nabla \cdot \left(E\times B/\mu_{0}\right).$$
Again, given that the Poynting flux divergence vanishes under a triply periodic volume average, we have
(18)$$\frac{d}{dt}\langle B^{2}/2\mu_{0}\rangle =-\langle E\cdot \nabla \times B\rangle /\mu_{0}.$$
Note here the difference to Equation (Equation 5), which ignores the displacement current. An equation similar to Equation (Equation 5) can only be recovered for the sum of electric and magnetic energies, which yields
(19)$$\frac{d}{dt}\left(\langle B^{2}/2\mu_{0}\rangle +\langle \epsilon_{0}E^{2}/2\rangle \right)=-\langle J\cdot E\rangle .$$
An important property of well-conducting media that are considered in magnetohydrodynamics is that the electric energy is negligible compared to the magnetic energy. In that limit, Equations (Equation 5) and (Equation 19) do indeed become equivalent.

More compactly, we can then write Equation (Equation 18) in the form ${\dot{E}}_{M}=-Q_{E}$, where $Q_{E}=\langle E\cdot \nabla \times B\rangle$ acts as a source in ${\dot{E}}_{E}=Q_{E}-\epsilon_{M}-W_{L}$. Thus, we clearly see that the electric energy reservoir is not a secondary one with an energy content that is small because of inefficient coupling, but it is an unavoidable intermediate one through which magnetic energy is channeled efficiently further to kinetic and thermal energies. This raises the question how safe is the neglect of the displacement current when prior to the emergence of conductivity, the electric energy dominates over magnetic energy. This is a typical situation in inflationary magnetohydrodynamic scenarios that we consider later in this paper. Before that, we first discuss the nonconducting case where electric and magnetic energy densities are equally large.

## 3. Numerical Experiments with Different Temporal Conductivity Variations

To illustrate the conversion from electromagnetic energy to magnetohydrodynamic and thermal energies during the emergence of electric conductivity, let us consider here a simple one-dimensional experiment.

### 3.1. Electromagnetic Waves and Their Suppression by Conductivity

In one dimension with ∂/∂x≠0 and σ=0, we can have electromagnetic waves, for example, $B_{y}^{\pm }\left(x,t\right)=B_{0}\sin k\left(c\mp ct\right)$ and $E_{z}^{\pm }\left(x,t\right)=\mp kB_{0}\sin k\left(c\mp ct\right)$, traveling in the positive (negative) *x* direction. Note that the electric and magnetic energies are here equal to each other. However, when σ becomes large, |E| becomes suppressed. To understand this suppression, let us look at Equation (Equation 1). When σ becomes large, the ∇×B term no longer needs to be balanced by the displacement current but by the actual current. Inserting J=σE (for the comoving current density), we find E=η∇×B, so we expect |E|/|cB|=O(ηk/c). Thus, once σ becomes large, |E|/c becomes suppressed relative to |B| by a factor, $\eta k/c=k/(c\mu_{0}\sigma )$. Additionally, as mentioned above, both |E| and |B| become suppressed due to the intermediate phase when σ is neither small nor large yet. This was discussed in the appendix of Ref. [8], who found that for a linearly increasing conductivity profile $\sigma \left(t\right)=\sigma_{\max}\;t/t_{\mathrm{trans}}$ during a certain time interval $t_{0}\leq t\leq t_{0}+t_{\mathrm{trans}}$ of duration $t_{\mathrm{trans}}$ and starting at $t=t_{0}$, there was an amplitude drop, whose value increases approximately inversely proportional to $\eta_{\min}k^{2}t_{\mathrm{trans}}$, where $\eta_{\min}=1/\mu_{0}\sigma_{\max}$.

To specify the temporal variation of the conductivity profile, we define a piecewise linear function that goes from 0 (for $t\leq t_{0}$) to 1 (for $t\geq t_{0}+\tau_{0}$) via
(20)$$\Theta =\max\left[\min\left(\frac{t-t_{0}}{\tau_{0}},\;1\right),\;0\right].$$

The linear σ profile used in Ref. [8] is given via
(21)$$\sigma \left(t\right)=\sigma_{\min}+\left(\sigma_{0}-\sigma_{\min}\right)\;\Theta \left(t\right),$$
where $\sigma_{0}=1/\mu_{0}\eta_{0}$ and $\sigma_{\min}=1/\mu_{0}\eta_{\max}$. Here, we also study a profile whose logarithm is linearly varying. We, therefore, refer to it as a logarithmic profile, which is of the form
(22)$$\sigma \left(t\right)=\sigma_{0}\exp\left\{\ln\left(\sigma_{\max}/\sigma_{0}\right)\;\left[1-\Theta (t)\right]\right\},$$
which allows us to specify the duration, during which σ transits by an order of magnitude for any value of σ. For the linear σ profile, in contrast, the duration would be different for different σ ranges, and it would be very short for large values of σ.

For the simulations, we use the Pencil Code [25], which employs the magnetic vector potential A, so that B=∇×A is always divergence free. The evolution equation for A can then be written as
(23)$$\frac{1}{c^{2}}\frac{\partial^{2}A}{\partial t^{2}}-\nabla^{2}A+\nabla \nabla \cdot A+\frac{1}{\eta (t)}\left(\frac{\partial A}{\partial t}+u\times B\right)=0,$$
where ∇·A=0 if the Coulomb gauge is used. In the Pencil Code, different gauges are possible, but the Weyl gauge with ∂A/∂t=−E is the one used in that code. In that case, the ∇∇·A term must be retained. Equation (Equation 23) shows that in a vacuum, where 1/η→0, one recovers a standard wave equation for waves with propagation speed *c*. In the opposite limit, where η→0, one can neglect the $\left(\eta /c^{2}\right)\;\partial^{2}A/\partial t^{2}$ term and one recovers the usual induction equation, where $\eta \nabla^{2}A$ acts as a diffusion term.

### 3.2. Transition to the High-Conductivity Regime for Different Parameters

The transition to the high-conductivity regime involves the conversion of electromagnetic waves to magnetohydrodynamic waves [9]. One can imagine that this process is more efficient when the frequencies of both waves are equal. In the high conductivity regime, the frequency of magnetohydrodynamic waves depends on the strength of the imposed magnetic field, $B_{0}$, which determines the nominal Alfvén speed, $v_{A0}=B_{0}/\sqrt{\rho \mu_{0}}$. Since Alfvén waves propagate along the magnetic field and since ∂/∂x≠0, we impose the magnetic field also in the *x* direction, i.e., we write $B=\hat{x}B_{0}+\nabla \times A$, where ∇×A is the departure from the imposed magnetic field. In the following numerical experiments, we choose $t_{0}=0$.

The results are shown in Figure 4, where we compare $B_{y}\left(x,t\right)$ as a colored contour plot in the xt plane. We compute the solution in a domain of size *L*, so the lowest wave number is $k_{1}=2\pi /L$. The density is initially uniform and equal to $\rho_{0}$. In the following, we use units where $c=k_{1}=\rho_{0}=1$. We see that for $v_{A0}=1$, the wave propagates almost unaffectedly by the switch to high conductivity. Here, the frequency of the electromagnetic wave is $ck_{1}=1$, and the nominal frequency of the Alfvén wave, $v_{A0}k$, is also unity, but the actual frequency is slightly less than that. This is because of the special relativity effects forcing the wave speed to be always less than *c*. In fact, the actual wave speed is $v_{A}=v_{A0}/{\left(1+v_{A0}^{2}/c^{2}\right)}^{1/2}$ [26].

For smaller values of $v_{A0}$, we see that not only the wave speed is less, as shown in the shallower inclination of the pattern in Figure 4b,c, but there is also a certain drop of the wave amplitude, and there is also an additional modulation resulting from an effective initial condition at t=0, which does not match the eigenfunction for an Alfvén wave.

In Figure 5, we compare the logarithmic σ profile with the linear one using $v_{A0}=0.3$. For the logarithmic profile, the drop in amplitude is clearly larger than that for the linear σ profile. To obtain a similar drop with the linear σ profile, one would need to increase $t_{\mathrm{trans}}$ to about 500; see Figure 5c.

To be more quantitative, we compare in Figure 6a the evolution of $B_{y}$ at one specific point $x=x_{*}$ for the three runs of Figure 5a,c. Note that the drop of the wave amplitude after t=0 is similar for runs a and c but much less for run b.

In Figure 6b, we also show how σ varies. We do this by plotting the nondimensional resistivity
(24)R(t)≡η(t)k/c,
which decreases from $10^{4}$ to $5\times 10^{-4}$. We recall that it is also this ratio that we identified in the beginning of this section as the one that characterizes the value of |E|/|cB|. We see that most of the decay happens when it transits through unity. Owing to the logarithmic nature of the profile, the quantity R(t) spends a time interval of about $ckt_{\mathrm{trans}}=5$, while R(t) changes from 10 to 0.1. In contrast, for the linear profile, the time interval is virtually non-existent. For $t_{\mathrm{trans}}=500$, on the other hand, $ckt_{\mathrm{trans}}$ is similar to what led to the to a similar decay for the logarithmic profile. This is also confirmed by the inset of panel **b**, which shows that R(t) traverses unity by a margin of one order of magnitude for **a** and **c** but not for **b**. The results discussed above confirm that the relevant time interval is indeed that where R(t) is within an order of magnitude around unity.

Looking at Figure 7, we see that the electric energy was initially equal to the magnetic one, but as the conductivity increases, there is a rapid decline of electric energy ($\dot{E_{E}}<0$), and most of it dissipates thermally, while only a small fraction (<10% for $t_{\mathrm{trans}}=10$) is transferred to kinetic energy. It turns out that the mean magnetic and electric energy densities decay like $\exp(-\nu k^{2}t)$, i.e., without a factor of 2 in the exponent, reflecting, therefore, not a change in the kinetic energy but rather in the velocity, which enters via the work term $W_{L}$. Furthermore, the ratio is $E_{E}/E_{M}\approx 10$ for ${\mathrm{Pr}}_{M}=20$. Note that the oscillations in $E_{M}+E_{K}$ (orange lines) are compensated mostly entirely by those in $E_{E}$ (blue lines).

In Figure 8, we show the evolution of various energy fluxes. We see that the magnetic energy decays and gives off energy to the electric energy reservoir through the term $Q_{E}=\langle E\cdot \nabla \times B\rangle >0$. The magnetic heating is thus composed of the following terms:(25)$$\epsilon_{M}=-{\dot{E}}_{E}+Q_{E}-W_{L}.$$
For rapid transits, $t_{\mathrm{trans}}≲5$, $Q_{E}$ is small compared with $-{\dot{E}}_{E}$, so $\epsilon_{M}$ is mostly entirely the result of exhausting electric energy, i.e., ${\dot{E}}_{E}<0$. For longer transits, $t_{\mathrm{trans}}>10$, $Q_{E}\approx -{\dot{E}}_{E}$, so $\epsilon_{M}$ is supplied to about 50% via $Q_{E}$ and to another 50% via $-{\dot{E}}_{E}$. These differences are summarized in Table 1.

In connection with Figure 4, we noted that there is a certain drop of the wave amplitude after the transit to large conductivity. This drop was larger for a larger ratio of the electromagnetic to Alfvén wave speeds. When the nominal Alfvén speed was equal to the speed of light, the drop was small. In Figure 9, we quantify this by plotting $E_{M}$ at t=100, i.e., after the conductivity has increased to a large value vs. $t_{\mathrm{trans}}$ for different values of $v_{A0}/c$. We confirm the results of Ref. [8], where the logarithmic drop was found to depend linearly on the value of $t_{\mathrm{trans}}$. However, we now also see that the slope of this curve decreases with increasing Alfvén wave speed.

## 4. Cosmological Application Prior to Radiation Domination

As alluded to above, the end of inflation might provide an opportunity to illustrate electromagnetic energy conversion because in that case, the electric energy can greatly exceed the magnetic one.

### 4.1. Magnetic Fields in Cosmology

In the present universe, magnetic fields are constantly being regenerated via dynamo action on all scales up to those of galaxy clusters. The energy source here is gravitational, which is released via accretion or direct collapse. Magnetic fields may also be present on even larger scales. However, in the locations between galaxy clusters, i.e., in what is often referred to as voids, it is generally thought impossible to produce magnetic fields via contemporary dynamo action; see Refs. [27,28] for reviews on the subject. Nevertheless, indirect evidence for the existence of magnetic fields in voids, and, more specifically, for lower limits of the magnetic field strength, comes from the non-observation of secondary photons in the halos to blazars, which are active galactic nuclei producing TeV photons. These photons interact with those of the extragalactic background light via inverse Compton scattering to produce GeV photons. Those secondary GeV photons are not observed. Their non-observation could be explained as an intervening magnetic field of about $10^{-16}\;G$ on a megaparsec scale [29,30]. This field would deflect electrons and positrons in opposite directions, preventing them from recombining and thereby disrupting the energy cascade toward the lower GeV photons.

The non-observation of GeV photons might have other reasons, for example, plasma instabilities that disrupt the electron–positron beam [31,32]. Nevertheless, even then, a certain fraction of the plasma beam disruption might still be caused by magnetic fields [33], which could explain the GeV halos of at least some blazars [34]. If magnetic fields really do exist on very large cosmological scales, they may be primordial in origin. This may mean that they have been created during or before the radiation-dominated era of the universe, for example, during one of the cosmological phase transitions or during inflation. Inflation is a phase where the conversion from electromagnetic fields to magnetohydrodynamic fields played an important role, which is what we are interested in here.

### 4.2. Use of Comoving Variables and Conformal Time

The universe expands with time, as described by the scale factor a(t). The equations of magnetohydrodynamics, therefore, contain additional terms with the factors of a(t) and its time derivatives. However, by using scaled variables, $\tilde{A}=aA$, $\tilde{B}=a^{2}B$, $\tilde{E}=a^{2}E$, $\tilde{J}=a^{3}J$, $\tilde{x}=x/a$, along with conformal time, $\tilde{t}=\int dt/a\left(t\right)$, all a(t) factors and other terms involving a(t) disappear from the magnetohydrodynamic equations [10]. The velocity is the same in both frames, i.e., $\tilde{u}=u$.

Given that the equations with tilded variables are equal to the ordinary ones in a non-expanded universe, it is convenient to skip all tildes from now on. However, when discussing the evolution of the scale factor, for example, we again need physical time, which will then be denoted as $t_{\mathrm{phys}}$, while *t* then still denotes conformal time. Here is where we have a notational dilemma because in cosmology, derivatives with respect to physical (or cosmic) time are often denoted by dots, while those with respect to conformal time are denoted by primes. We, therefore, decided here to follow the same convention, so $a^{\prime }=da/dt$ and $a^{\prime \prime }=d{}^{2}a/dt^{2}$ denote derivatives with respect to conformal time.

### 4.3. Inflationary Magnetogenesis

Inflationary magnetogenesis models assume the breaking of conformal invariance through a coupling to a scalar field such as the inflaton. Another possible coupling is through an axion field, which would result in helical magnetogenesis, but this will not be considered here. The dynamics of the scalar field is interesting in its own right; see Refs. [35,36,37] for numerical investigations. To simplify the model, one commonly replaces this coupling by a prefactor $f^{2}$, where *f* depends on the scale factor of the universe. This factor $f^{2}$ enters in the electromagnetic energy contribution to the Lagrangian density $f^{2}F^{\mu \nu }F_{\mu \nu }$, where $F_{\mu \nu }$ is the Faraday tensor [38]. Early approaches to inflationary magnetogenesis exposed specific problems: the strong coupling and the backreaction problems [39], as well as the Schwinger effect constraint, which can lead to a premature increase in the electric conductivity. This shorts the electric field and prevents further magnetic field growth [20]. This is particularly important for models that solve the backreaction problem by choosing a low energy scale inflation [40] but could be avoided if charged particles attain sufficiently large masses via some mechanism in the early universe [41]. The three problems are avoided by requiring the function *f* to obey certain constraints [7,42].

Successful models of inflationary magnetogenesis are thus possible, but this does not mean that the underlying cosmological models are also physically preferred options. Nevertheless, for the purpose of discussing the electromagnetic energy conversion, which is the goal of this paper, those models are a useful choice.

Three-dimensional simulations of inflationary magnetogenesis have been performed by assuming an abrupt switch from electromagnetism without currents and magnetohydrodynamics where the displacement current is already neglected [8,43]. They solved the evolution equations for the scaled magnetic vector potential, A≡fA, in the Coulomb gauge: (26)$$\left[\frac{1}{c^{2}}\frac{\partial^{2}}{\partial t^{2}}-\nabla^{2}-k_{*}^{2}\left(t\right)\right]A=0.$$
where $k_{*}^{2}\left(t\right)=f^{\prime \prime }/f$ is a generation term because it destabilizes the field at large length scales for wavenumbers $k<k_{*}\left(t\right)$. Analogous to the primes on a(t), primes on f(t) also denote conformal time derivatives. Toward the end of the reheating phase, where f→1, we expect $k_{*}\left(t\right)\to 0$.

Our aim here is to present calculations where the transit from vacuum to high conductivity is continuous. In particular, to calculate the generation term $k_{*}^{2}\left(t\right)$, one commonly uses a power law representation in terms of a(t) of the form $f\propto a^{\alpha }$ with α>0 during inflation and $f\propto a^{-\beta }$ with β>0 during reheating [44]. We are here only interested in the reheating phase, where $a\left(t\right)\propto t^{2}$ [7,42], such that it is unity when the radiation-dominated era begins, and therefore, f=1 and $k_{*}^{2}\left(t\right)=0$ for a>1. For a<1, in contrast, we have
(27)$$k_{*}^{2}\left(t\right)=\beta \left[\left(\beta +1\right){\left(a^{\prime }/a\right)}^{2}-a^{\prime \prime }/a\right].$$
Note that for $a=t^{2}$, we have $a^{\prime }=2t$ and $a^{\prime \prime }=2$, so ${\left(a^{\prime }/a\right)}^{2}=4/t^{2}$ and $a^{\prime \prime }/a=2/t^{2}$, and therefore, $f^{\prime \prime }/f=2\beta \left(2\beta +1\right)/t^{2}$.

Contrary to the earlier numerical work [8,43], the displacement current is now included at all times. However, there is still a problem in that $k_{*}^{2}\left(t\right)$ has a discontinuity from $k_{*}^{2}\left(1\right)=\beta \left(\beta +1\right)\neq 0$ to zero at the moment when the conductivity is turned on. In the simulations, this did not seem to have any serious effect on the results because the magnetic field at the end of the electromagnetic phase only acted as an initial condition for the magnetohydrodynamic calculation after the switch. In a continuous calculation without switch, however, this problem must be avoided. This will be addressed next.

### 4.4. Continuous Version of the Generation Term

An instructive way of obtaining a smooth transition from a quadratic to a linear growth profile of a(t) is obtained by solving the Friedmann equations for a piecewise constant equation of state, w(a), which relates the pressure with the density through p=wρ. Under the assumption of zero curvature, i.e., the Universe is conformally flat, but expanding, the Friedmann equations can be written as a single equation, which, in physical time, takes the form of
(28)$$a^{-1}\;d{}^{2}a/dt_{\mathrm{phys}}^{2}=-\frac{1}{2}H^{2}\left[1+3w(a)\right],$$
where $H=a^{-1}da/dt_{\mathrm{phys}}$ is the standard Hubble parameter. Here, w(a)=1/3 during the radiation-dominated era and w(a)=0 during reheating when there were no photons, which is, therefore, equivalent to the matter-dominated era that also occurs later after recombination and before the universe began to accelerate again. The accelerated exponential expansion of the universe during inflation, and also the late acceleration of the present universe, correspond to w=−1, but this will not be considered in this present paper.

It is convenient to solve the Friedmann equation with zero curvature in conformal time. It then takes the form $a^{\prime \prime }/a=\frac{1}{2}H^{2}\;\left(1-3w\right)$, where $H=a^{\prime }/a$ is the conformal Hubble parameter. It is related to the usual one, *H*, via $H=da/\partial t_{\mathrm{phys}}=aH$. Note the opposite sign of the terms on the right-hand side and the opposite sign in front of 3w(a) compared to the formulation in terms of physical time. The equation for $a^{\prime \prime }$ is easily solved by splitting it into two first-order equations and introducing a new variable b(t) and solving for
(29)$$a^{\prime }=b,\;b^{\prime }=\left(b^{2}/2a\right)\left(1-3w\right);$$
see also Ref. [45] for similar work in another context.

Figure 10 shows the solution for a(t) and the ratios $a^{\prime }/a$ and $a^{\prime \prime }/a$ compensated by *t* and $t^{2}$, respectively, which allows us to see more clearly how $a^{\prime }/a$ changes from 2/t to 1/t and $a^{\prime \prime }/a$ changes from $2/t^{2}$ to zero as we go from the reheating era to the radiation-dominated universe after reheating.

The generation term $k_{*}^{2}\left(t\right)\equiv f^{\prime \prime }/f$ determines the wavenumber below, of which the solution is still unstable. However, since $k_{*}^{2}\left(t\right)=2\beta \left(2\beta +1\right)/t^{2}$, we have ctk=const≈2β+1/2; see Table 2. In Figure 11, we plot the evolution of $E_{E}$, $E_{M}$, and $E_{K}$ for k=10 for all three values of β: 1, 2, and 4. Here and below, the initial amplitudes have been arranged such that $E_{M}=10^{-4}$ at $t=t_{0}$. In all cases, the solution has become stable by the time $t=t_{0}=1$, and we see electromagnetic oscillations toward the end of the reheating phase before conductivity turns on at $t_{0}=1$. This is here referred to as Set (i).

It is easy to see that on large length scales, when the $\nabla^{2}$ operator in Equation (Equation 26) is negligible compared with $k_{*}^{2}\left(t\right)$, we have
(30)$$\begin{array}{cc} & A_{z}\left(x,t\right)=A_{0}t^{2\beta +1}k^{-1}\cos kx,\;A_{z}\left(x,t\right)=A_{z}/f=A_{0}t^{\beta +1}k^{-1}\cos kx, \end{array}$$
(31)$$\begin{array}{cc} & B_{y}\left(x,t\right)=A_{0}t^{\beta +1}\sin kx,\;E_{z}\left(x,t\right)=-\partial A_{z}/\partial t=-\left(\beta +1\right)A_{0}t^{\beta }k^{-1}\cos kx. \end{array}$$
Thus, for ckt≪1, corresponding to the super-horizon scales, where and when the modes are still unstable, we have $tE_{\mathrm{rms}}/B_{\mathrm{rms}}\approx \beta +1$. On smaller length scales, i.e., for larger *k* values, the modes become stable, and we have the usual electromagnetic waves.

When modeling the transition from a vacuum to that of high conductivity and the corresponding Joule heating, we still need to make a choice as to when σ would begin to increase, i.e., we need to choose values of $t_{0}$ and $t_{\mathrm{trans}}$. If we choose the value of $t_{0}$ to be too large, we obtain solutions where electromagnetic waves have already been established; see Figure 11. The smallest wavenumber in our one-dimensional domain is k=10, so by the time t=1, even the largest modes in the domain are stable. We also see that at early times, $E_{E}$ and $E_{M}$ grow in an algebraic fashion and then become oscillatory when $k_{*}\left(t\right)$ has dropped below *k*. At $t=t_{0}=1$, when conductivity turns on, the electric energy decreases rapidly, while the magnetic energy diminishes only very slowly. The generated hydrodynamic energy is, however, small. This is similar to what we studied in Section 3.2.

Our objective here is to study cases that are different from what was studied in Section 3.2. Therefore, we now choose Set (ii) with $t_{0}=0.1$ and $t_{\mathrm{trans}}=1$ (Figure 12) and another Set (iii) with $k_{1}=1$ and $t_{0}=1$ (Figure 13). Again, the electric energy drops significantly when conductivity turns on, but now there is a much larger spread in the resulting maximum magnetic energies for the three cases with β=1, 2, and 4. For β=4, $E_{K}$ reaches about one percent of $E_{M}$ at t=0.2, for example.

When increasing the wavenumber to k=1, the largest modes are still unstable for the three cases with β=1, 2, and 4; see Figure 13. Here, k=1 and $t_{0}=1$ and $t_{\mathrm{trans}}=10$. The spread in the magnetic energy is similar, but the maximum kinetic energy is now much larger; see Table 3.

### 4.5. Energy Conversions during Reheating

During reheating, there is an additional source of energy resulting from the generation term $k_{*}^{2}\left(t\right)$. The term $k_{*}^{2}\left(t\right)$ appeared in Equation (Equation 26) for A=fA. However, to write down the relevant equation for E=−∂A/∂t, we have to revert to the original equation for A, which reads [44]
(32)$$\left[\frac{1}{c^{2}}\left(\frac{\partial^{2}}{\partial t^{2}}+2\frac{f^{\prime }}{f}\frac{\partial }{\partial t}\right)-\nabla^{2}\right]A=0.$$
Thus, Equation (Equation 1) with the current density term restored, now becomes
(33)$$\frac{1}{c^{2}}\left(\frac{\partial E}{\partial t}+2\frac{f^{\prime }}{f}E\right)=\nabla \times B-\mu_{0}J,$$
and therefore, Equation (Equation 15) for the electric energy now has an extra term and reads
(34)$$\frac{d}{dt}\langle \epsilon_{0}E^{2}/2\rangle =-2\left(f^{\prime }/f\right)\langle \epsilon_{0}E^{2}\rangle +\langle E\cdot \nabla \times B/\mu_{0}\rangle -\langle J\cdot E\rangle .$$
During reheating with $f\propto a^{-\beta }\propto t^{-2\beta }$, we have $f^{\prime }/f=-2\beta /t$, so the first term on the right-hand side of Equation (Equation 34) is positive for β>0, so there is growth of the electric energy. Similarly to what was performed in Section 3.2, we can write the electric energy equation more compactly as ${\dot{E}}_{E}=Q_{G}+Q_{E}-\epsilon_{M}-W_{L}$, where $Q_{G}=-4\left(f^{\prime }/f\right)E_{E}$ is now the dominant source, but here, $Q_{E}$ plays the role of a sink during the first part of the evolution. This equation generalizes Equation (Equation 25) to the case with electromagnetic field generation during reheating; see also Figure 14. Unlike the earlier case of Figure 3, where there was no energy input, we here have a system that it driven by energy input through the $Q_{G}$ term.

The evolution of $Q_{G}$, $Q_{E}$, ${\dot{E}}_{E}$, and $\epsilon_{M}$ is shown in Figure 15 during magnetic field generation in case (ii) for all three values of β. It is instructive to write the electric energy equation as
(35)$$Q_{G}=\epsilon_{M}+{\dot{E}}_{E}+W_{L}-Q_{E}.$$
Comparing the three panels of Figure 15, we thus see that for β=1, there is a slow generation phase starting much before $t_{0}$. It should be noted, however, that the ranges on the vertical axes are different for the different panels.

At $t=t_{0}=0.1$, there is a rise of conductivity and, therefore, a sharp rise in Ohmic heating, $\epsilon_{M}$. This is also the time when ${\dot{E}}_{E}$ reaches a maximum and becomes negative shortly thereafter. For large values of β, this moment happens a bit later, at t=0.11 compared to t=0.10 for β=1. Note that, while for β=4, the maxima of $Q_{G}$ and $\epsilon_{M}$ are similar, for the smaller values of β, the maxima of $\epsilon_{M}$ are much larger than those of $Q_{G}$. Instead, for β=1, for example, we have $\epsilon_{M}\approx -{\dot{E}}_{E}$, i.e., almost the entire heating is here caused by dissipation of electric energy.

In Figure 16, we show a plot similar to Figure 15, but for the case (i), where all modes were already oscillatory at $t=t_{0}=1$, when conductivity turned on. The Ohmic heating now plays a minor role in the sense that its maximum value is much less than the extrema of $Q_{G}$, $Q_{E}$, and ${\dot{E}}_{E}$. For β=1 and 2, we see that ${\dot{E}}_{E}$ and $Q_{E}$ are nearly in phase shortly before conductivity turns on. This means that the electric and magnetic energies are strongly coupled, and a flow of energy from magnetic to electric energy ($Q_{E}>0$) leads to an increase in electric energy (${\dot{E}}_{E}>0$). This is expected because there is only an oscillatory exchange between electric and magnetic energies. For β=4, on the other hand, the oscillatory phase just started to develop shortly before t=1, but the curves are similar to those for β=1 and 2, although they shifted toward earlier times. The time of the first maximum of $Q_{G}$ is at t=0.9 for β=4, while for β=2, it is at t=0.46, and for β=1, it is at t=0.25, and we see that the profiles of all curves are indeed very similar around those times.

## 5. Conclusions

In this paper, we have studied the conversion of electromagnetic energy into kinetic and thermal energies as the electric conductivity transits from zero (vacuum) to large values. This problem has relevance to the reheating phase at the end of cosmological inflation and before the emergence of a relatively long radiation-dominated era before the time of recombination, which is much later (on a logarithmic time scale). While not much is known about the physical processes leading to reheating and the emergence of conductivity, a lot can now be said about the general process of such an energy conversion.

Already in the absence of cosmological expansion, we have seen that the transition to conductivity involves an oscillatory exchange between electric and magnetic energies. It is mainly the electric energy reservoir that delivers energy to the kinetic and thermal energy reservoirs and not the magnetic energy directly as in magnetohydrodynamics. We knew already from earlier works that the duration of the transit plays a significant role in causing a drop in magnetic energy. We now also see that this drop depends on the magnetic field strength and, thus, the typical Alfvén speed. The drop can become small if the Alfvén speed becomes comparable to the speed of light. Furthermore, for short transits, we have seen that the energy transfer between electric and magnetic energies is small and that the initial electric energy goes directly into thermal energy. For longer transits, however, the mutual exchange with magnetic energy becomes approximately equal to the thermal energy loss, so thermalization now also involves the magnetic energy reservoir.

When applying electromagnetic energy conservation to the problem of reheating, we have a new quality in the model in that there is now also energy transfer through conformal invariance breaking, which may occur during inflation and reheating. This is obviously speculative [27,28,38,46,47,48,49,50] but a very promising scenario for the generation of large-scale magnetic fields in the early universe and for explaining the observed lower limits of the intergalactic magnetic field on megaparsec length scales [29,30].

This present study has shown that significant work can be done by the Lorentz force when the electromagnetic energy conversion happens early and on scales large enough so that the modes are still growing in time. This is because there is then a significant excess of electric energy over magnetic. This is an effect that was ignored in previous simulations of inflationary magnetogenesis and, in particular, in studies of the additional contributions to the resulting relic gravitational wave production [8,43].

It will be useful to extend our studies to turbulent flows and magnetic fields. This requires that one solves for the evolution of $\rho_{e}$ and that $\nabla \cdot E=\rho_{e}/\epsilon_{0}$ is obeyed at all times. This constraint was automatically obeyed in our one-dimensional models. In the future, it would also be interesting to study dynamo action in situations of moderate magnetic conductivity where coupling with the electric energy reservoir could reveal new aspects.

## Figures and Tables

**Figure 1 entropy-25-01270-f001:**
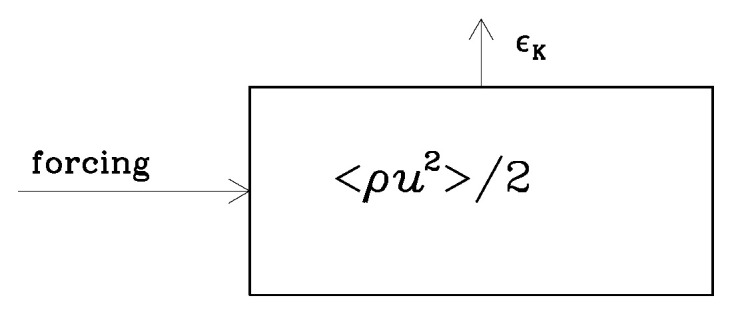
Kinetic energy dissipation, $\epsilon_{K}$, in forced turbulence with kinetic energy density $\langle \rho u^{2}\rangle /2$, where ρ is the density, and u is the velocity. In the steady state, $\epsilon_{K}$ equals energy input via forcing.

**Figure 2 entropy-25-01270-f002:**
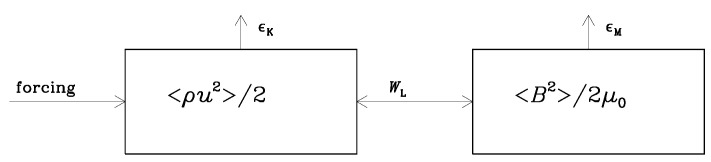
Dissipation in dynamos. There are now two exit channels, $\epsilon_{K}$ and $\epsilon_{M}$, and it is not clear who takes the lion’s share. Dynamo action corresponds to $W_{L}<0$ (i.e., the work conducted against the Lorentz force, which is indicated by the arrow on $W_{L}$ pointing to the right), although energy can also go the other way around when the magnetic field is strong and drives motions via the Lorentz force.

**Figure 3 entropy-25-01270-f003:**
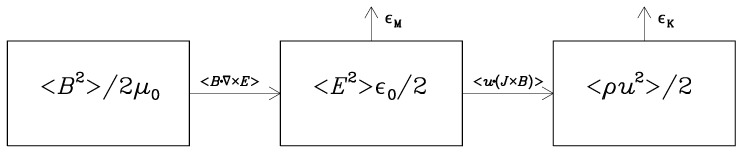
Energy conversion from magnetic to kinetic energies via the electric energy reservoir.

**Figure 4 entropy-25-01270-f004:**
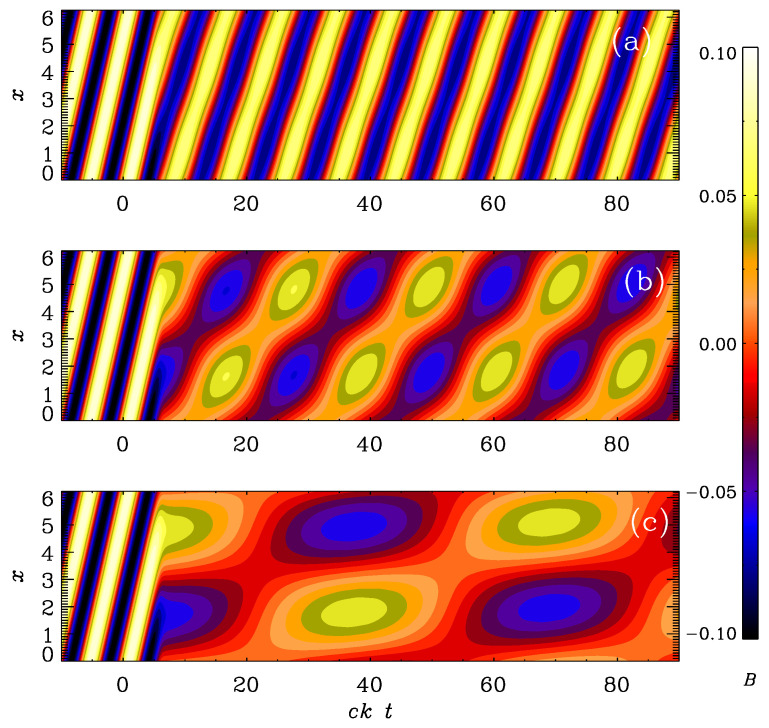
Evolution of $B_{y}\left(x,t\right)$ for the logarithmic σ profile with (**a**) $v_{A0}=1$, (**b**) $v_{A0}=0.3$, and (**c**) $v_{A0}=0.1$, and $t_{\mathrm{trans}}=10$ in all cases.

**Figure 5 entropy-25-01270-f005:**
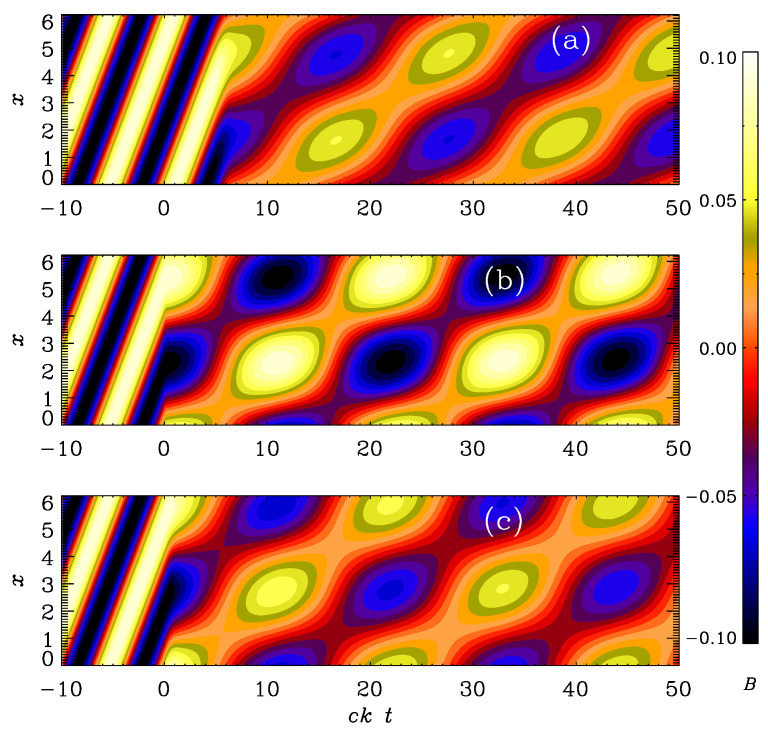
Evolution of $B_{y}\left(x,t\right)$ for the logarithmic σ profile with $v_{A}=0.3$ and (**a**) the logarithmic σ profile with $t_{\mathrm{trans}}=10$, (**b**) the linear σ profile with $t_{\mathrm{trans}}=10$, and (**c**) the linear σ profile with $t_{\mathrm{trans}}=500$.

**Figure 6 entropy-25-01270-f006:**
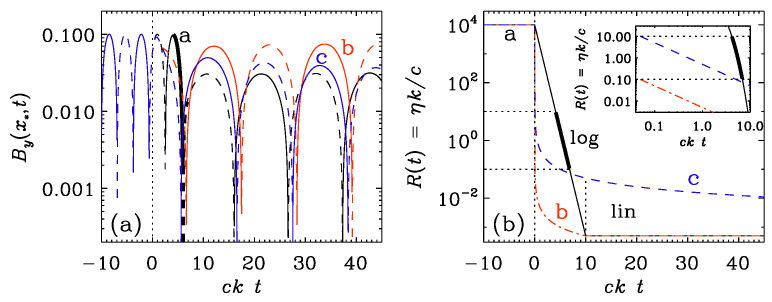
(**a**) Evolution of $B_{y}$ at one specific point $x=x_{*}$ in the three runs of Figure 5a,c. Note that the drop of the wave amplitude after t=0, and specifically at t=50, is similar for runs a and c, but much less for **b**. (**b**) Dependence of the nondimensional resistivity R(t)=η(t)k/c for the logarithmic profile with $t_{\mathrm{trans}}=10$ in run a (black), and the linear profile with $t_{\mathrm{trans}}=10$ in run b (red) and $t_{\mathrm{trans}}=500$ in run c (blue). The inset shows a blow-up of a narrow strip around R=1 using a logarithmic time axis. We see from the inset that the time spent in R(t) traversing unity by a margin of one order of magnitude (marked by the thick part of the black line) is similar for runs a and c, but virtually non-existing for run b.

**Figure 7 entropy-25-01270-f007:**
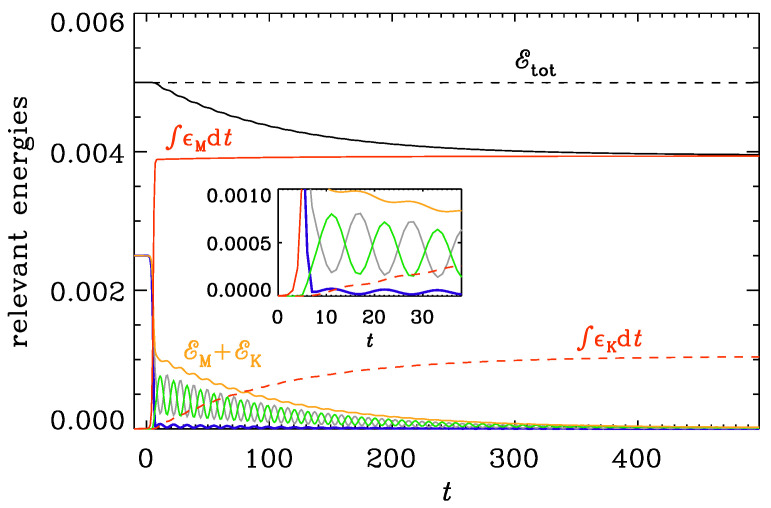
Initially, all the energy is in electromagnetic energy, $E_{E}+E_{M}$ for ν=0.01 and $\eta_{\mathrm{fin}}=5\times 10^{-4}$. In the end, all the energy is converted into heat. The red lines give the integrated Ohmic and viscous energy gains, $\int \epsilon_{M}\;dt$ and $\int \epsilon_{K}\;dt$, respectively. At intermediate times, this energy is distributed to equal amounts among kinetic energy $E_{K}$ (green lines) and magnetic energy $E_{M}$ (gray lines). The orange lines shows their sum, $E_{K}+E_{M}$. The blue lines represent $E_{E}$. The inset shows a blow-up of the same graph around the origin. We see that $E_{E}$ varies in phase with $E_{K}$, but an anti-phase both with $E_{M}$ and the residual $E_{M}+E_{K}$.

**Figure 8 entropy-25-01270-f008:**
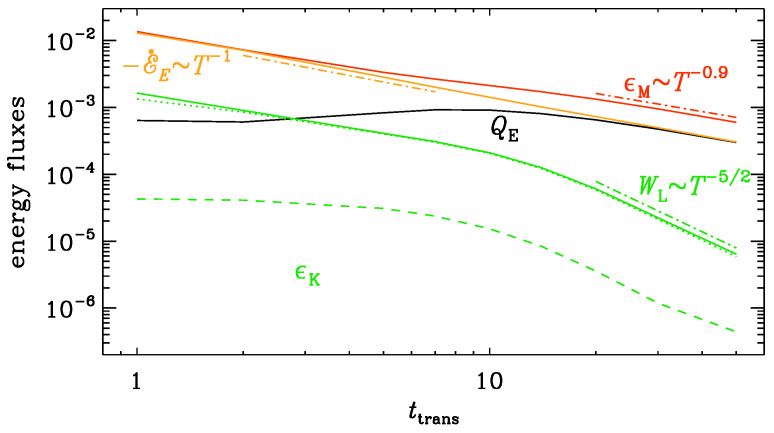
Evolution of energy fluxes for the model with the logarithmic conductivity profile with $\eta =5\times 10^{-4}=\nu$ at late times. In all cases, the initial diffusivity is $\eta_{\mathrm{ini}}=10^{4}$. The main difference to the run with a larger viscosity is that $\epsilon_{K}$ is larger.

**Figure 9 entropy-25-01270-f009:**
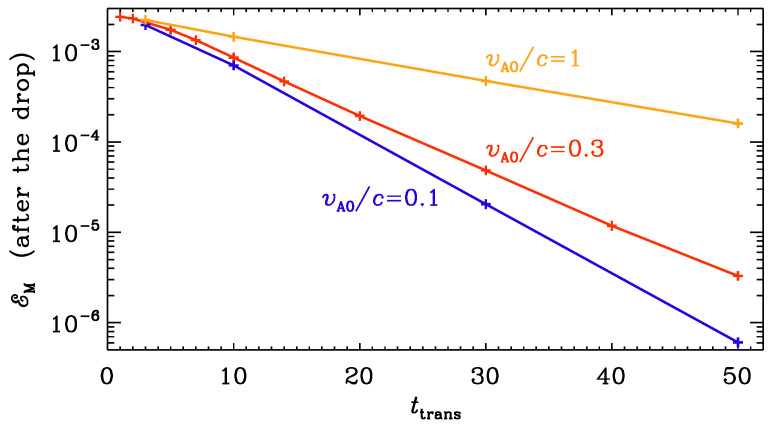
$E_{M}$ at t=100, i.e., after the conductivity has increased to large value, vs. $t_{\mathrm{trans}}$ for $v_{A0}/c=1$ (orange), $v_{A0}/c=0.3$ (red), and $v_{A0}/c=0.1$ (blue).

**Figure 10 entropy-25-01270-f010:**
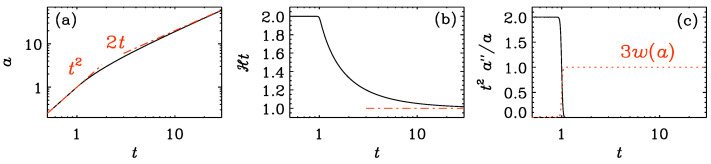
*t* dependence of (**a**) the scale factor a(t), (**b**) the compensated Hubble coefficient $Ht=t\;a^{\prime }/a$, and (**c**) the compensated left-hand side of the Friedmann equation, $t^{2}\;a^{\prime \prime }/a$. In (**a**), the asymptotic dependences $a=t^{2}$ and 2t for t≪1 and ≫1 are overplotted as dashed-dotted orange lines. In (**c**), the function 3w(a(t)) is overplotted as a dotted red line.

**Figure 11 entropy-25-01270-f011:**
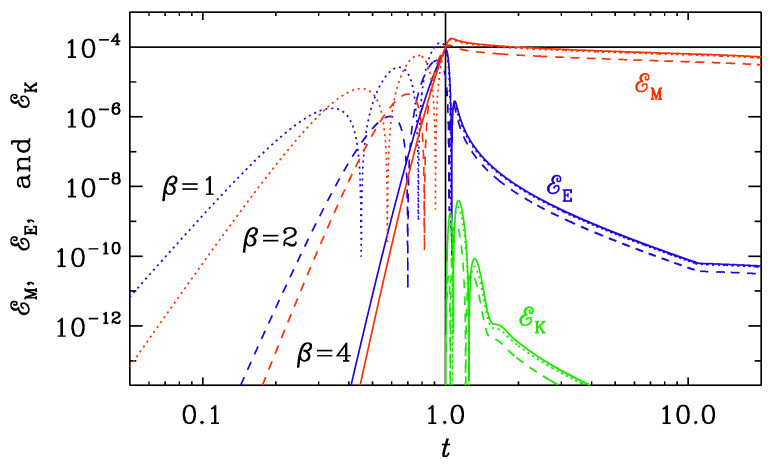
*t* dependence of $E_{M}$ (red), $E_{E}$ (blue), and $E_{K}$ (green) for runs with β=1 (dotted lines), 2 (dashed lines), and 4 (solid lines) for Set (i) with k=10, $t_{0}=1$, and $t_{\mathrm{trans}}=10$. The initial amplitudes have been arranged such that $B_{\mathrm{rms}}=0.01$ at t=1. From the double-logarithmic representation, we see that the growth of $E_{M}$ and $E_{E}$ is algebraic, and much faster for the models with a larger value of β. Before t=1, $E_{E}$ dominates over $E_{M}$, but drops immediately after t=1, when resistivity emerges and kinetic energy is being generated. Both $E_{M}$ and $E_{K}$ are larger for larger values of β.

**Figure 12 entropy-25-01270-f012:**
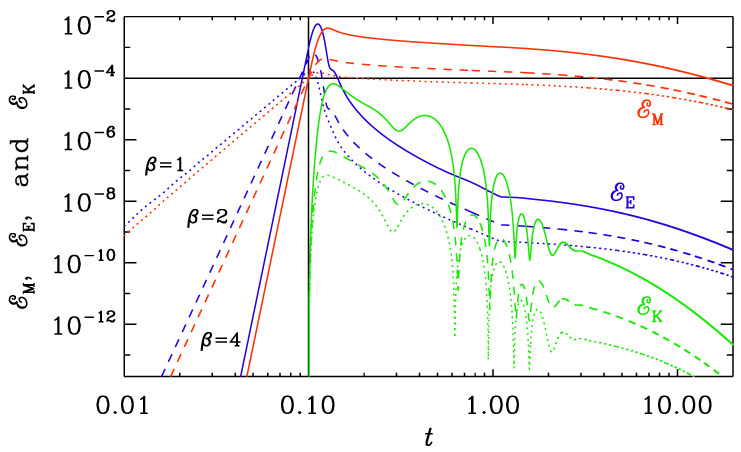
Similarly to Figure 11, but for Set (ii) with k=10, $t_{0}=0.1$, and $t_{\mathrm{trans}}=1$.

**Figure 13 entropy-25-01270-f013:**
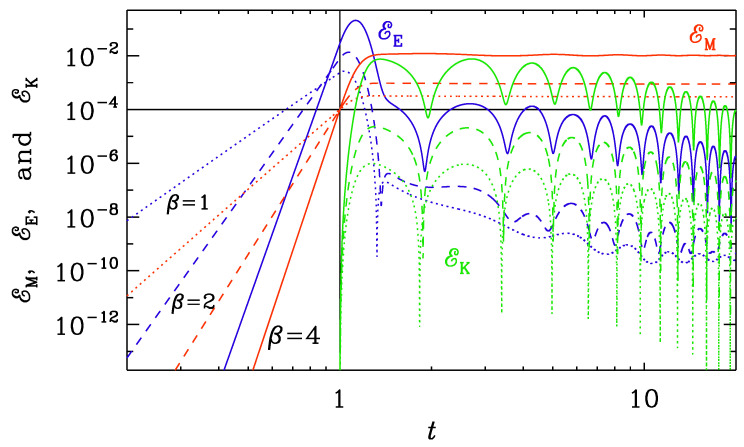
Similarly to Figure 11, but for Set (iii) with k=1, $t_{0}=1$, and $t_{\mathrm{trans}}=10$.

**Figure 14 entropy-25-01270-f014:**
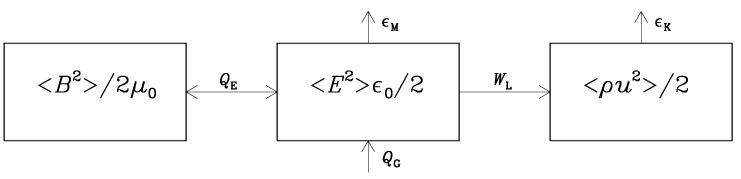
Similar to Figure 3, but now with inflationary magnetogenesis energy generation and energy exchange between electric and magnetic energies in both directions.

**Figure 15 entropy-25-01270-f015:**
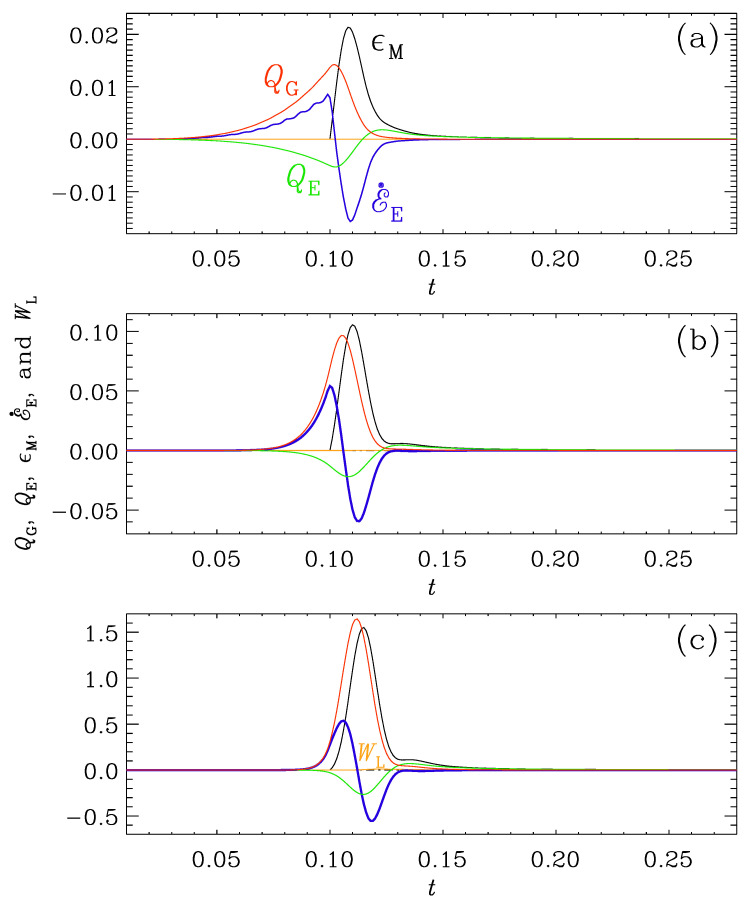
*t* dependence of $Q_{G}$ (red), $Q_{E}$ (green), $\epsilon_{M}$ (black), ${\dot{E}}_{E}$ (blue), and $W_{L}$ (orange) for the runs of Set (ii) in Figure 12 with (**a**) β=1, (**b**) β=2, and (**c**) β=4 for $t_{0}=0.1$ and $t_{\mathrm{trans}}=1$.

**Figure 16 entropy-25-01270-f016:**
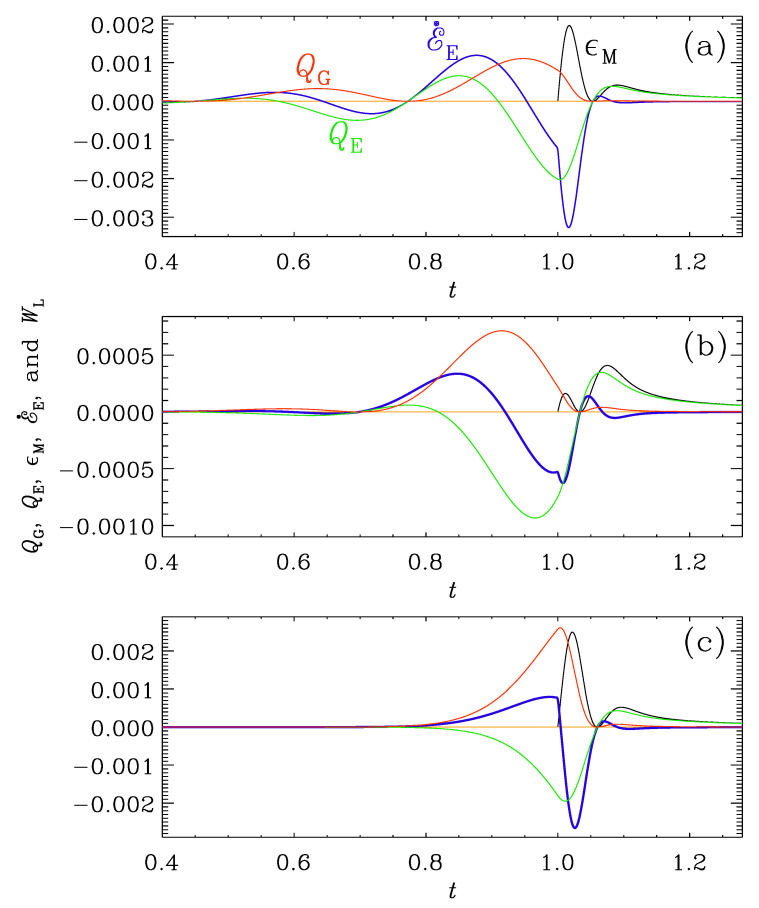
Similarly to Figure 15, but for the runs of Set (i) in Figure 11 with $t_{0}=1$ and $t_{\mathrm{trans}}=10$, and (**a**) β=1, (**b**) β=2, and (**c**) β=4. Note that $Q_{G}$ and $Q_{E}$ vary in anti-phase.

**Table 1 entropy-25-01270-t001:** The two regimes of energy transfer for short and long transits.

	Rapid Transits	Long Transits
criterion	$t_{\mathrm{trans}}<10$	$t_{\mathrm{trans}}>10$
Lorentz work	$W_{L}/\epsilon_{M}\approx 0.1$	$W_{L}/\epsilon_{M}\approx 5\;t_{\mathrm{trans}}^{-1.6}$
heating	$\epsilon_{M}\approx -{\dot{E}}_{E}$ and $Q_{E}\ll \epsilon_{M}$	$\epsilon_{M}\approx -0.5\;{\dot{E}}_{E}\approx 0.5\;Q_{E}$

**Table 2 entropy-25-01270-t002:** Parameters relevant for the models with different values of β.

β	2β+1	2β+1/2	$k_{*}\left(1\right)$
1	3	2.5	2.45
2	5	4.5	4.47
4	9	8.5	8.49

**Table 3 entropy-25-01270-t003:** Summary of various extrema for each of the three sets of models and values of β.

Set	*k*	$t_{0}$	Variable	β=1	β=2	β=4
(i)	10	1	$\max E_{E}$	$1.3\times 10^{-4}$	$4.1\times 10^{-5}$	$8.3\times 10^{-5}$
(ii)	10	0.1		$1.8\times 10^{-4}$	$6.4\times 10^{-4}$	$5.8\times 10^{-3}$
(iii)	1	1		$2.7\times 10^{-3}$	$1.4\times 10^{-2}$	$2.1\times 10^{-1}$
(i)	10	1	$\max E_{M}$	$1.6\times 10^{-4}$	$1.1\times 10^{-4}$	$1.8\times 10^{-4}$
(ii)	10	0.1		$1.5\times 10^{-4}$	$4.3\times 10^{-4}$	$4.2\times 10^{-3}$
(iii)	1	1		$3.2\times 10^{-4}$	$9.6\times 10^{-4}$	$1.2\times 10^{-2}$
(i)	10	1	$\max E_{K}$	$2.5\times 10^{-9}$	$9.0\times 10^{-10}$	$3.9\times 10^{-9}$
(ii)	10	0.1		$6.9\times 10^{-8}$	$4.2\times 10^{-7}$	$6.6\times 10^{-5}$
(iii)	1	1		$9.8\times 10^{-7}$	$2.3\times 10^{-5}$	$7.7\times 10^{-3}$
(i)	10	1	$\max Q_{G}$	$1.4\times 10^{-4}$	$2.0\times 10^{-4}$	$1.4\times 10^{-3}$
(ii)	10	0.1		$4.4\times 10^{-3}$	$3.8\times 10^{-2}$	$8.6\times 10^{-1}$
(iii)	1	1		$8.4\times 10^{-3}$	$7.8\times 10^{-2}$	$2.7\times 10^{0}$
(i)	10	1	$\max(-Q_{E})$	$2.0\times 10^{-3}$	$9.3\times 10^{-4}$	$2.0\times 10^{-3}$
(ii)	10	0.1		$5.3\times 10^{-3}$	$2.2\times 10^{-2}$	$2.7\times 10^{-1}$
(iii)	1	1		$1.3\times 10^{-3}$	$4.7\times 10^{-3}$	$5.9\times 10^{-2}$
(i)	10	1	$\max\epsilon_{M}$	$2.0\times 10^{-3}$	$4.1\times 10^{-4}$	$2.5\times 10^{-3}$
(ii)	10	0.1		$2.1\times 10^{-2}$	$1.1\times 10^{-1}$	$1.5\times 10^{0}$
(iii)	1	1		$3.5\times 10^{-2}$	$2.5\times 10^{-1}$	$5.8\times 10^{0}$
(i)	10	1	$\max(-{\dot{E}}_{E})$	$3.3\times 10^{-3}$	$6.3\times 10^{-4}$	$2.7\times 10^{-3}$
(ii)	10	0.1		$1.6\times 10^{-2}$	$5.9\times 10^{-2}$	$5.6\times 10^{-1}$
(iii)	1	1		$2.4\times 10^{-2}$	$1.3\times 10^{-1}$	$1.9\times 10^{0}$
(i)	10	1	$\max{\dot{E}}_{E}$	$1.2\times 10^{-3}$	$3.4\times 10^{-4}$	$8.0\times 10^{-4}$
(ii)	10	0.1		$8.6\times 10^{-3}$	$5.4\times 10^{-2}$	$5.4\times 10^{-1}$
(iii)	1	1		$1.8\times 10^{-2}$	$1.3\times 10^{-1}$	$2.0\times 10^{0}$

## Data Availability

The source code used for the simulations of this study, the Pencil Code [25], is freely available on https://github.com/pencil-code/ (accessed on 1 August 2023). The DOI of the code is https://doi.org/10.5281/zenodo.2315093 (accessed on 1 August 2023). The simulation setups and the corresponding secondary data are available on http://norlx65.nordita.org/~brandenb/projects/EMconversion (accessed on 1 August 2023) and on https://doi.org/10.5281/zenodo.8203242 (accessed on 1 August 2023).

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
