# Peer review of "Electromagnetic Conversion into Kinetic and Thermal Energies"

_entropy, 2023, doi:10.3390/e25091270_

Round 1
Reviewer 1 Report
This is an extremely interesting and original idea regarding the origin of cosmic magnetic fields. However it should be emphasized at the beginning that the idea is extremely speculative. The introductory manipulations of Maxwell's equation s are not very original but the interpretations are and perhaps are questionable.
The biggest single omission at the beginning of the paper is the failure to account for non-closed systems. This allows the authors to omit all reference to the theoretical and observational studies of galactic magnetic fields associated with the CHANG-ES. Their theoretical multi-scale models require open systems and the predictions show satisfactory agreement with radio and dust observations.
It is true that the application to the problem of the origin of magnetic field in the Universe does not encounter this problem, but the cost is the extra speculation.
More specifically:
1. Equation (1) seems to be missing an epsilon_o. Moreover this equation is often referred to as equation (32) as after equation (14) and in sections 3 and 4.5.
2. The notion of `triply periodic' deserves a brief explanation.
3. In the discussion after (7) the formW_L= u.J^B doesn't have a lot to do with the classical dynamo. It can also be written as -J.(u^B) which is E_(induced).J and is the work of the induced electric field on the current. If it takes the third form -B.(J^u), it looks like the classical dynamo term provided that <J^u> is independent of B. Some of the statements in this section seem a bit arbitrary. Shouldn't epsilon_m be a time derivative?
4. In equation (11) is d/dt not equal to D/Dt?
5. It seems that equations (15) and (18) ignore equations (5) and (7).
6. The numerical studies in one dimension are of interest. It is certainly not new to use the vector potential. However, why is it in figures 4,5 the value of ckt_trans is about the same? Figures 6 and 7 are too complicated and need more specific captions.
7. Section 4 is extremely speculative. The possible meaning of f(a) is not discussed. figures 11 and similar are complicated and could use clearer descriptions. Why does the Lorentz force not relax to zero at very low densities? This would give a simple equation for B, possibly time dependent during the expansion. u would have to parallel B as well. Extensive experimentation with non-linear, scale invariant force free fields is available in Henriksen , R N, 2019.
Author Response
We thank the two referees for their comments and suggestions. We have made
corresponding changes that are marked in blue.
Response to Referee 1:
> This is an extremely interesting and original idea regarding the origin of
> cosmic magnetic fields. However it should be emphasized at the beginning
> that the idea is extremely speculative. The introductory manipulations
> of Maxwell's equation s are not very original but the interpretations
> are and perhaps are questionable.
As we also mention below, inflationary magnetogenesis has received
significant attention in the scientific community, and we have now added
many more citations at the end of that sentence in the conclusions; see
paragraph 3, line 4, on page 19.
> The biggest single omission at the beginning of the paper is the failure
> to account for non-closed systems. This allows the authors to omit all
> reference to the theoretical and observational studies of galactic
> magnetic fields associated with the CHANG-ES. Their theoretical
> multi-scale models require open systems and the predictions show
> satisfactory agreement with radio and dust observations.
We understand the concern of the referee. In Figs. 1 and 2, we had
explicit energy input through forcing. We have highlighted this now in
the new paragraph after Eq.(10). The energy input during inflation that
is discussed in Section 4.5 is also a case with energy input. We have
mentioned this now in this new paragraph. We also mention this briefly
at the end of the paragraph after Eq.(34) on page 17. The referee also
alludes to other astrophysical applications, but in those other cases,
the electric energy is negligible. This is now highlighted more explicitly
on page 2, in addition to what is said later in section 3.1 on page 6.
We have now also added some sentences at the end of the paragraph on Silk
damping in the middle of page 2 that electric energies are negligible
in most of astrophysics and cosmology. Galaxies and the interstellar
medium, and even the voids are regions where the conductivity is high,
so electric energy is negligible. A more precise condition has been
presented in the beginning of Section 3.1.
> It is true that the application to the problem of the origin of magnetic
> field in the Universe does not encounter this problem, but the cost is
> the extra speculation.
We ourselves used the word "speculative", but to emphasize that
inflationary magnetogenesis has received significant attention in the
scientific community, we have now added many more references at the end
of that sentence in the conclusions; see paragraph 3, line 4, on page
19. The word speculative should therefore not be understood as something
specific to this particular work.
> More specifically:
> 1. Equation (1) seems to be missing an epsilon_o. Moreover this equation
> is often referred to as equation (32) as after equation (14) and in
> sections 3 and 4.5.
Equation (1) should be correct in its current form. To arrive at an
equation for the electric energy, one divides by mu_0, so we have
1/(mu_0*c^2), which is equal to epsilon_0, which is what the referee
was probably thinking of.
Regarding the reference to Eq.(32), we thank the referee for having
pointed out our mistake with a replicated latex label, which has now
been corrected.
> 2. The notion of `triply periodic' deserves a brief explanation.
After Eq.(4) on page 3, we have now added some sentences explaining
the use of volume averaging, the angle brackets, and also the use of
periodic boundary conditions in all three directions, which we refer to
as triply periodic.
> 3. In the discussion after (7) the form W_L= u.J^B doesn't have a lot
> to do with the classical dynamo. It can also be written as -J.(u^B)
> which is E_(induced).J and is the work of the induced electric field
> on the current. If it takes the third form -B.(J^u), it looks like the
> classical dynamo term provided that <J^u> is independent of B. Some of
> the statements in this section seem a bit arbitrary. Shouldn't epsilon_m
> be a time derivative?
As we stated ourselves after Equation (6), all the three permutations
are equivalent and correspond to a coupling between magnetic and kinetic
energy reservoirs. Dynamo action corresponds to energy flow from kinetic
to magnetic, i.e., when W_L is negative. This should be true of all
selfexcited dynamos. There are also non-selfexcited dynamos, which are
discussed in magnetospheric physics, but not usually in astrophysics.
Regarding epsilon_m being a time derivative, there is potentially a
confusion between a calligraphic E, which is the magnetic energy density
if it has the subscript M. Yes, its negative time derivative is partially
balanced by epsilon_M, i.e., the magnetic dissipation, but this epsilon
has no time derivative. In the paragraph after Eq.(19), the calligraphic
E_M does have a dot on top, although it can be had to see.
> 4. In equation (11) is d/dt not equal to D/Dt?
No, because angular brackets denote volume averages, so they don't
depend on spatial coordinates. To clarify this, we have now also added
"They depend just on $t$, but not on x." in the text after Eq.(4).
> 5. It seems that equations (15) and (18) ignore equations (5) and (7).
Equation (5) should be *contrasted* with Eqs.(15) and (18), because
the former *neglects* the displacement current. We have now stated this
explicitly with reference to the new Eq.(19).
> 6. The numerical studies in one dimension are of interest. It is
> certainly not new to use the vector potential. However, why is it in
> figures 4,5 the value of ckt_trans is about the same? Figures 6 and 7
> are too complicated and need more specific captions.
In Fig.4, we change vA0/c and keep ckt_trans=10=const, while in Fig.5,
we change the profiles and compare ckt_trans=10 and 500. To do a more
quantitative comparison with different values of ckt_trans, we have
Figures 8 and 9, but don't show images any more. Yes, Figures 6 and 7
are rather complex, and the yellow line was hard to see. We have now
changed that into a gray one and added an inset to see more clearly
the curves near the origin. We have now also expanded the captions of
figures 6 and 7.
> 7. Section 4 is extremely speculative. The possible meaning of f(a)
> is not discussed. figures 11 and similar are complicated and could use
> clearer descriptions. Why does the Lorentz force not relax to zero at
> very low densities? This would give a simple equation for B, possibly
> time dependent during the expansion. u would have to parallel B as
> well. Extensive experimentation with non-linear, scale invariant force
> free fields is available in Henriksen, R N, 2019.
We have now expanded the caption of Fig.11. The Lorentz force could
diminish in response to the resulting flows, but it would not usually
relax to zero, unless the current helicity is finite. We have now referred
to the possibility a vanishing Lorentz force in the text after Eq.(7). We
have now explained in more detail the meaning of f(a) in the beginning
of Sect.4.3 on page 12.
Reviewer 2 Report
This paper is a clearly written review of electromagnetic issues to be addressed in understanding the early Universe at the end of inflation. Our knowledge of this environment is limited but this paper considers basic physical ideas relevant to the sharing of various forms of electromagnetic and kinetic energies under a suite of useful assumptions.
The paper clearly presents the physics which is relevant to the study and the early introductory sections would be good reading for many students of electromagnetic processes in conducting environments. Later parts of the paper deal with physics in assumed ranges of properties in a regime of Universe expansion which is poorly quantified, or even described, in the literature.
The presentation is clear and exemplary. I noted some small issues:
line 92: did the authors mean Eqn (32). I may have missed the argument but this didn't seem what was intended.
line 104 and Equation (12). Not an issue for the authors but the (important) dot over the kinetic energy was close to invisible in the displayed manuscript.
line 136: 'safe in' => 'safe is'
line 208: 5(b) => 6(b) I believe
Figure 7 has a yellow line which is almost impossible to see. Can the color be changed?
The Conclusions (final paragraph) note limitations due to the neglect of turbulence in flows. Some comment of the likely effect of this would be welcome.
Author Response
We thank the two referees for their comments and suggestions. We have made
corresponding changes that are marked in blue.
Response to Referee 2:
> This paper is a clearly written review of electromagnetic issues to be
> addressed in understanding the early Universe at the end of inflation.
> Our knowledge of this environment is limited but this paper considers
> basic physical ideas relevant to the sharing of various forms of
> electromagnetic and kinetic energies under a suite of useful assumptions.
> The paper clearly presents the physics which is relevant to the study and
> the early introductory sections would be good reading for many students
> of electromagnetic processes in conducting environments. Later parts of
> the paper deal with physics in assumed ranges of properties in a regime
> of Universe expansion which is poorly quantified, or even described,
> in the literature.
> The presentation is clear and exemplary. I noted some small issues:
We thank the reviewer for their positive and constructive assessment.
> line 92: did the authors mean Eqn (32). I may have missed the argument
> but this didn't seem what was intended.
Regarding the reference to Eq.(32), we did mean Eq.(1). We thank the
referee for having pointed out our mistake with a replicated latex label,
which has now been corrected.
> line 104 and Equation (12). Not an issue for the authors but the
> (important) dot over the kinetic energy was close to invisible in the
> displayed manuscript.
The Latex fonts used by the publisher are bit unfortunate in that respect.
We have now added a sentence after Eq.(12) to highligh the presence of
the dot.
> line 136: 'safe in' => 'safe is'
We have now corrected this.
> line 208: 5(b) => 6(b), I believe
Yes, this was a mistake and we have now corrected this; see page 8,
just before Eq.(24).
> Figure 7 has a yellow line which is almost impossible to see.
> Can the color be changed?
We have now changed the yellow line into a gray one and added an inset
to see more clearly the curves near the origin. We have now also enhanced
the caption.
> The Conclusions (final paragraph) note limitations due to the neglect
> of turbulence in flows. Some comment of the likely effect of this would
> be welcome.
We did already explain the numerical difficulties we have encountered
with more complex flows. We hope we can overcome them in future. Of
course, it would be of great interest to consider turbulent flows.
Round 2
Reviewer 1 Report
I read their response and was satisfied.